# Extravasation and fluid collection on computed tomography imaging in patients with colonic diverticular bleeding

Hitomi Takada[1,2]* *, Makoto Kadokura[1,2]*, Tomoki Yasumura[1,2], Hiroki Yoda[1,2], Tetsuya Okuwaki[1,2], Naoto Imagawa[2], Naruki Shimamura[2], Keisuke Tanaka[1,2], Fumitake Amemiya[1,2], Nobuyuki Enomoto[2]

1 Department of Gastroenterology and Hepatology, Municipal Hospital of Kofu, Yamanashi, Japan, 2 First Department of Internal Medicine, Faculty of Medicine, University of Yamanashi, Yamanashi, Japan

* These authors contributed equally to this work.
* takadahi0107@gmail.com

**Data Availability Statement:** All relevant data are within the paper.

**Funding:** The authors received no specific funding for this work.

## Abstract

### Objective

We evaluated the characteristics of patients with diverticular bleeding in whom emergency endoscopy should be proactively performed and those in whom it is unnecessary for spontaneous hemostasis following conservative treatment.

### Methods

This study involved 132 patients in whom diverticular bleeding was diagnosed on lower gastrointestinal endoscopy. We evaluated the rate of identification of the bleeding diverticulum during endoscopy and the rate of spontaneous hemostasis following conservative treatment.

### Results

In 26 patients (20%), bleeding diverticulum was identified during endoscopy. Extravasation or fluid collection on CT imaging was an important factor of successful identification of the bleeding source on endoscopy. Of the 104 patients in the conservative treatment group, 91 (87%) were able to be discharged after spontaneous hemostasis. Univariate analysis revealed a high rate of spontaneous hemostasis in patients without extravasation and fluid collection on CT imaging, those without adhesion of blood during endoscopy, those without diabetes, and those with a hemoglobin level $\geq$10 g/dL.

### Conclusion

In patients with colonic diverticular bleeding, extravasation or fluid collection on CT is an important factor related to the identification of the bleeding diverticulum. Patients without characteristic CT findings had a high rate of spontaneous hemostasis after conservative treatment.

**Competing interests:** The authors have declared that no competing interests exist.

## Background

Diverticular bleeding is the most frequent cause of lower gastrointestinal bleeding accounting for 20%–40% of all cases in Japan and 20%–48% of all those in the Western countries [1, 2]. The prevalence of colonic diverticula tends to increase with age; thus, the overall prevalence of diverticular bleeding is expected to increase in the future. In Japan, the Japanese Gastroenterological Association published guidelines on colonic diverticulitis in 2017; these guidelines recommend the performance of lower gastrointestinal endoscopic examination within 24 h in patients with lower gastrointestinal bleeding suspected to be diverticular bleeding[3]. It has been reported that, for patients with lower gastrointestinal bleeding, urgent endoscopy helps avoid embolotherapy, colectomy, massive blood transfusion, and repeat bleeding[1, 4, 5]. However, it is often difficult to identify the bleeding point [6]; further, there are many challenging cases wherein it is difficult to decide whether urgent endoscopy should be performed in situations where there is insufficient medical staff, such as during nighttime and on holidays. Bleeding is reported to stop spontaneously with conservative treatment alone in 70% of diverticular bleeding cases[7, 8]. In particular, when determining the treatment policy for diverticular bleeding and in the case of patients at high risk of complications following endoscopy, such as older patients, those with poor performance status or cardiovascular disease, and those in whom spontaneous hemostasis can be expected, urgent endoscopy should be avoided, and elective endoscopy should be selected. Therefore, the type of cases wherein urgent endoscopy is effective and the type wherein it is unnecessary need to be clarified. Thus far, there have been very few reports of the characteristics of patients with diverticular bleeding in whom spontaneous hemostasis was achieved.

We aimed to assess the characteristics of patients in whom emergency endoscopy should be proactively performed and those for whom it is unnecessary. Thus, we retrospectively analyzed the identification rate for the responsible diverticulum in patients with diverticular bleeding and the rate of spontaneous hemostasis following conservative treatment.

## Methods

### Statement of ethics

This study was conducted as per the principles of the Helsinki Declaration and the ethical guidelines for epidemiological research presented by the Ministry of Health, Labour and Welfare in Japan[9, 10]. The institutional review board of Municipal Hospital of Kofu approved the study.

### Subjects

The subjects included 132 patients who visited Municipal Hospital of Kofu with a complaint of bloody stools from 2011 to 2019 and in whom diverticular bleeding was diagnosed on urgent or elective lower gastrointestinal endoscopy. We excluded patients who had experienced the last bloody stool at least one week previously, those who received treatment for diverticular bleeding within one month before the consultation, and those in whom the source of bleeding was not the diverticulum. The following data was collected from these patients' records and

retrospectively analyzed: age, sex, medical history, oral medications, vital signs, physical findings, blood test findings, and imaging examination findings. This study was conducted as per the principles of the Helsinki Declaration and the ethical guidelines for epidemiological research presented by the Ministry of Health, Labour and Welfare in Japan. The institutional review board of Municipal Hospital of Kofu approved the study (ethics committee for clinical studies of Municipal Hospital of Kofu: Rinshoukenkyu-Rinrishinsa-Iinkai (in Japanese), approval number 31–24), the ethics committee waived the requirement for informed consent. All data were fully anonymized before we accessed them.

## Diagnosis and treatment of diverticular bleeding

After confirming the stabilization of vital signs, physical examination and blood testing were performed. In patients without allergy to the contrast agent and renal failure, contrast-enhanced computed tomography (CT) was proactively performed. For the contrast-enhanced CT, 90 mL of iopamidol was intravenously administered at a rate of 1.5 mL/s, and the scan was taken using a 16- or 64-slice detector CT. Urgent or elective lower gastrointestinal endoscopy was performed at the discretion of the attending physician based on the vital parameters, medical examination results, and CT findings. Lower gastrointestinal endoscopy was performed by two physicians, including an endoscopy specialist and a non-specialist, using a PCF-H290I (Olympus, Tokyo, Japan) device, and a tip hood and water-jet system (Olympus) were concurrently used at the discretion of the physicians. Pretreatment for elective endoscopy was performed in all patients using magnesium citrate or polyethylene glycol. The presence or absence of pretreatment at the time of urgent endoscopy was determined at the discretion of the physician. On endoscopic examination, the diverticula with the stigmata of recent hemorrhage (SRH) exhibiting active bleeding, exposed vessels, or adhesion of blood clots were diagnosed as the bleeding source. Endoscopic clipping performed using the direct method or the sewing method. For patients in whom endoscopic hemostasis was difficult, additional embolotherapy or colectomy was performed.

## Analysis

Statistical analyses were performed using Fisher's test for categorical variables and an unpaired student's t-test for continuous variables. A p value < 0.05 was considered statistically significant difference. An intergroup comparison was performed using log-rank test. Results are presented as median (range) values and numbers (%). All the statistical analyses were performed using EZR (Saitama Medical Center, Jichi Medical University, Saitama, Japan), a graphical user interface for R (The R Foundation for Statistical Computing, Vienna, Austria). More precisely, it is a modified version of the R commander designed to include the statistical functions that are frequently used in biostatistics[11].

## Results

### Patient background

The patients' background information is presented in Table 1. The average patient age was 72 (36–96) years, and the sample included 81 male patients (61%). Among those, 24 patients (18%) had a history of diverticular bleeding, 37 (28%) were taking oral antithrombotic agents, and 8 (6.0%) were taking nonsteroidal anti-inflammatory drugs (NSAIDs). At the time of consultation, the shock index was 0.62 (0.31–1.65), systolic blood pressure was 134 (78–206) mmHg, and hemoglobin level was 12 (5.1–17) g/dL. CT was performed at the time of consultation in 97 patients (73%), and contrast-enhanced CT was performed in 75 patients (57%).

**Table 1. Backgrounds of patients with diverticular bleeding.**

|  | n = 132 |
| --- | --- |
| age: years | 74 (36–96) |
| Male: n (%) | 81 (61%) |
| Body mass index | 23 (15–38) |
| Past history of diverticular bleeding: n (%) | 24 (18%) |
| Diabetes mellitus: n (%) | 20 (17%) |
| Hypertension: n (%) | 55 (46%) |
| Ischemic heart disease: n (%) | 21 (17%) |
| Cerebrovascular disease: n (%) | 13 (16%) |
| Chronic renal failure: n (%) | 9 (7.4%) |
| Use of antithrombic drugs: n (%) | 37 (28%) |
| Use of 2 or more antithrombic drugs: n (%) | 14 (12%) |
| Use of NSAIDs: n (%) | 8 (6.0%) |
| Systolic blood pressure at admission: mmHg | 134 (78–206) |
| Diastolic blood pressure at admission: mmHg | 80 (35–150) |
| Pulse at admission: /minute | 83 (39–186) |
| Shock index at admission | 0.62 (0.31–1.65) |
| White Blood Cell: $\times 10^3/\mu l$ | 6.8 (1.8–16) |
| Hemoglobin: g/dl | 12 (5.1–17) |
| Platelet: $\times 10^3/\mu l$ | 220 (58–400) |
| Albumin: g/dl | 3.9 (2.7–4.8) |
| estimate Glomerular Filtration Rate: ml/min/1.73m$^2$ | 62 (3.3–108) |
| C-Reactive Protein: mg/dl | 0.10 (0.0–14) |
| Prothrombin time: % | 99 (15–181) |
| Minimum hemoglobin: g/dl | 9.7 (6–16) |
| Implementation of CT at admission: n (%) | 97 (73%) |
| Extravasation on CT | 18 (19%) |
| Thickening of colon wall on CT | 20 (21%) |
| Fluid level on CT | 29 (30%) |
| Rise in concentration of surrounding fat tissue on CT | 15 (15%) |
| Extravasation or fluid level on CT | 33 (34%) |
| Time to the first colonoscopy: hours | 27 (1–195) |
| Time to the first colonoscopy within 24 hours: n (%) | 47 (36%) |
| Use of blood transfusion: n (%) | 48 (36%) |
| Recurrence bleeding: n (%) | 13 (13%) |
| Time to recurrent bleeding: days | 2 (1–32) |
| Hospitalization days: days | 9 (3–59) |

Data are expressed as median (range) or number (%).

CT: computed tomography, NSAIDs: nonsteroidal anti-inflammatory medications, SD: standard deviation.

Extravasation, fluid collection, bowel wall thickening, and increased density of the surrounding fat tissue was observed in 18 (19%), 29 (30%), 20 (21%), and 15 (15%) patients, respectively (Figs 1 and 2). Fluid collection represents high-attenuation on single phase CT images, and the attenuation value was 63 (51–118) Hounsfield Units[12]. Extravasation or fluid collection was observed in 33 patients (34%). Total 72% of the patients with extravasation and 23% without extravasation had fluid collection. The period from consultation until the endoscopy was 27 (1–195) hours, and urgent endoscopy within 24 h of the consultation was performed for 47

**Fig 1. Extravasation on CT.** (a) Plain CT image shows wall thickening in the ascending colon. (b) Arterial contrast material-enhanced CT image shows high attenuation area. (c) Delayed CT image reveals a larger area of attenuation. (d,e) Endoscopic examination shows active bleeding in the ascending colon.

patients (36%). Pretreatment at the time of emergency endoscopy within 24 h was performed for 19 patients (40%). Red blood cell transfusion was performed for 48 patients (36%). The mean hospital stay was 9.0 (3.0–59) d.

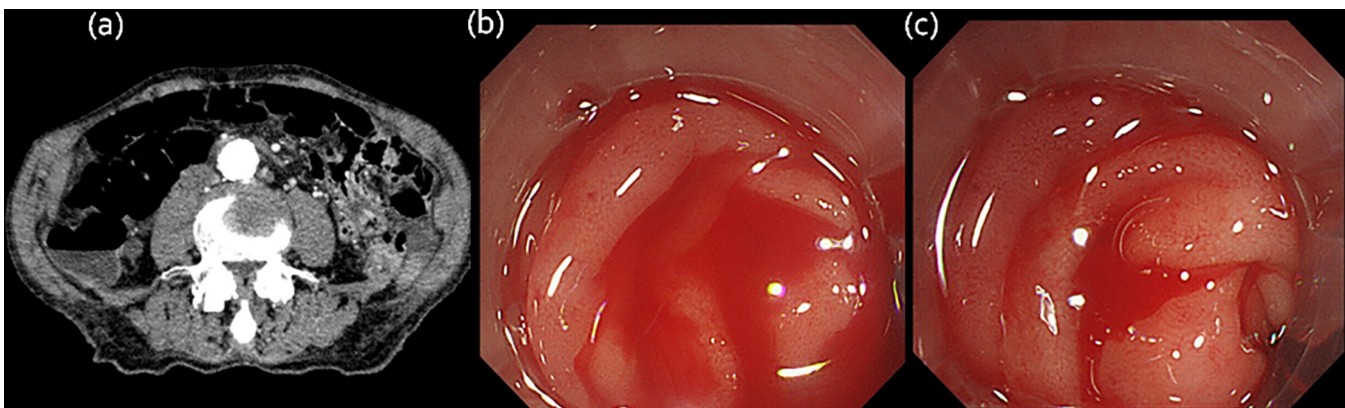

**Fig 2. Fluid level on CT.** (a) Plain CT image represents fluid collection showing high-attenuation in the ascending colon. (b,c) Endoscopic examination shows active bleeding in the ascending colon.

**Table 2. Univariate and Multivariate analysis of factors to detect the bleeding diverticulum on colonoscopy.**

| Factors | Univariate analysis | | | Multivariate analysis | | |
|---|---|---|---|---|---|---|
| | OR | 95% CI | P value | OR | 95% CI | P value |
| Age > 80: years old | 2.4 | 1.9–3.8 | 0.052 | | | |
| Men | 1.6 | 0.64–4.1 | 0.32 | | | |
| Diabetes mellitus | 1.7 | 0.45–6.3 | 0.44 | | | |
| Past history of diverticular bleeding | 1.6 | 0.29–9.1 | 0.58 | | | |
| Use of antithrombic drugs | 1.6 | 0.59–4.3 | 0.35 | | | |
| Use of NSAIDs | 1.3 | 0.24–6.6 | 0.79 | | | |
| Shock index at admission | 2.0 | 0.33–12 | 0.46 | | | |
| Hemoglobin at admission: g/dl | 1.2 | 0.95–1.4 | 0.15 | | | |
| Extravasation or fluid level on CT | 17 | 4.2–17 | <0.001 | 10 | 3.7–14 | <0.001 |
| Time to the first colonoscopy within 24 hours | 5.2 | 3.6–13 | <0.001 | 3.1 | 2.0–3.8 | 0.050 |

CT: computed tomography, NSAIDs: nonsteroidal anti-inflammatory medications, OR; Odds Ratio.

## The identification rate of the bleeding diverticulum during lower gastrointestinal endoscopy

In 26 patients (20%), the bleeding diverticulum was identified during endoscopy, and hemostasis with endoscopic clipping was performed for all these patients. Univariate analyses showed that the following factors were significantly associated with the identification of bleeding diverticulum: older age > 80 y (OR 2.4, p = 0.052), extravasation or fluid collection on CT imaging (OR 17, p < 0.001), and endoscopy performed within 24 h of consultation (OR 5.2, p = 0.0003). We found no difference in the oral medication, medical history, and bleeding site of the patients (Table 2). The 11 patients with bleeding findings in the absence of extravasation and fluid collection on CT imaging were more elderly compared to the patients without bleeding findings (83(64–89) vs. 74 (42–96) years old, p = 0.034), but no other background differences; sex, vital signs, laboratory data, oral medication, bleeding site, and other CT findings. Sensitivity, specificity, positive predictive value, negative predictive value, and accuracy to predict the identification of the bleeding diverticulum by CT imaging are shown in Table 3.

## The rate of spontaneous hemostasis following conservative treatment

The 106 patients who did not undergo endoscopic clipping included two patients in whom the procedure was changed to embolotherapy or surgery due to difficulty in endoscopic hemostasis attributed to a poor visual field caused by the massive bleeding. In residual 104 patients in

**Table 3. Sensitivity, specificity, positive predictive value, negative predictive value and accuracy to predict the identification of the bleeding diverticulum by CT imaging.**

| | Extravasation | Fluid collection | Extravasation or fluid collection |
|---|---|---|---|
| Sensitivity | 48% (26–70) * | 71% (48–89) * | 81% (58–95) * |
| Specificity | 90% (80–95) * | 79% (68–88) * | 75% (64–84) * |
| Positive predictive value | 56% (31–79) * | 48% (30–67) * | 47% (30–65) * |
| Negative predictive value | 86% (77–93) * | 91% (81–97) * | 93% (84–98) * |
| Accuracy | 80% (71–88) * | 77% (68–85) * | 76% (67–84) * |

CT: computed tomography.

* 95% CI: confidence interval.

**Table 4. Univariate analysis of factors for spontaneous hemostasis following conservative treatment.**

| Factors | OR | 95% CI | P value |
|---|---|---|---|
| Age | 0.98 | 0.94–1.0 | 0.39 |
| Men | 0.97 | 0.28–3.3 | 0.96 |
| Absence of Diabetes mellitus | 4.2 | 1.2–16 | 0.029 |
| Use of antithrombic drugs | 0.41 | 0.08–2.0 | 0.28 |
| Use of NSAIDs | 1.4 | 0.15–13 | 0.76 |
| Shock index at admission | 1.2 | 0.08–18 | 0.91 |
| Hemoglobin at admission: g/dl | 0.85 | 0.66–1.1 | 0.22 |
| Minimum hemoglobin more than 10: g/dl | 11 | 1.3–87 | 0.026 |
| Absence of extravasation and fluid level on CT | 6.0 | 1.5–24 | 0.010 |
| Time to the first colonoscopy within 24 hours | 2.6 | 0.76–8.9 | 0.13 |
| Absence of adhered blood on colonoscopy | 32 | 3.9–26 | 0.0012 |

CT: computed tomography, NSAIDs: nonsteroidal anti-inflammatory medications, OR; Odds Ratio.

whom SRH was not observed during endoscopy, a conservative treatment was adopted. Among the 104 patients in the conservative treatment group, 13 (13%) exhibited re-bleeding during hospitalization and required additional treatment. The average duration from admission until re-bleeding was 2 (1–32) d. The remaining 91 patients (87%) were able to be discharged after spontaneous hemostasis was achieved. As factors associated with spontaneous hemostasis following conservative treatment, univariate analysis revealed a high rate of spontaneous hemostasis in patients without extravasation and fluid collection on CT imaging (OR 6.0, p = 0.010), those without adhesion of blood during endoscopy (OR 32, p = 0.0012), those without diabetes (OR 4.2, p = 0.029), and those with hemoglobin level $\geq$10 g/ dL (OR 11, p = 0.026) (Table 4).

## Discussion

In the present study, we analyzed the relationship of imaging examination findings and the identification of the bleeding diverticulum and the characteristics of spontaneous hemostasis after conservative treatments in patients with diverticular bleeding. The identification rate of bleeding diverticulum was 20% with the performance of lower gastrointestinal endoscopy. The factor associated with bleeding diverticulum identification was extravasation or fluid collection on CT imaging. Colonic diverticular bleeding is 20%–42% of the cause of lower gastrointestinal bleeding[1]. It has been reported that patients in whom lower gastrointestinal endoscopy is performed within 12 h for lower gastrointestinal bleeding have a high rate of identification of the bleeding source (OR 2.6, CI 1.1–6.2), and the prognosis was good[6, 13, 14]. In Japan, the Japanese Gastroenterological Association published guidelines on colonic diverticulitis in 2017 that recommend performing lower gastrointestinal endoscopic examination within 24 h in patients with lower gastrointestinal bleeding suspected to be diverticular bleeding[3]. The American Society for Gastrointestinal Endoscopy guideline only recommends performing endoscopy early and does not stipulate a specific time[15]. Whether urgent endoscopy should be performed for all patients with lower gastrointestinal bleeding suspected of diverticular bleeding remains controversial. Adequate endoscopy staff is not always available to perform an urgent colonoscopy, and urgent examination should be selected and performed in patients for whom it is considered possible to identify the bleeding source and achieve hemostasis endoscopically. Furthermore, in patients in whom spontaneous hemostasis can be expected, the elective endoscopic examination should be selected. At present, there are several reports

regarding the identification rate for bleeding diverticulum and the re-bleeding rate[16]; however, to our knowledge, few reports describe the relationship of the identification rate of responsible diverticulum and CT findings (extravasation and fluid collection), and the characteristics of cases in which spontaneous hemostasis was achieved following conservative treatment, making the present study the first.

It has previously been reported that the bleeding diverticulum identification rate in diverticular bleeding is 10%–68%. In particular, the identification rate is higher among patients who undergo CT prior to endoscopic examination than in those who do not undergo CT (35.7 vs. 20.6%, p = 0.01). Furthermore, it has been reported that the bleeding diverticulum identification rate is higher in patients with extravasation observed on contrast-enhanced CT than in those without (68% vs. 20%, p < 0.001)[17, 18]. Therefore, all patients without allergy to the contrast medium and renal impairment undergo CT examination as the initial screening at our hospital. In the present study, 73% of the patients underwent a CT examination; this percentage is higher than that reported previously, 15%–55%[18]. The identification rate was significantly higher among patients who exhibited extravasation on CT than among those who did not show extravasation and those who did not undergo CT (56% vs. 16%, p < 0.001). The identification rate was significantly higher among patients with fluid collection on CT compared to those without and those who did not undergo CT (48% vs. 13%, p < 0.001). Ichiba et al. reported that among 257 patients with lower gastrointestinal bleeding suspected to be diverticular bleeding, the percentage of patients with fluid collection on CT was 32%. In their study, 40% subjects exhibited extravasation and 50.5% had endoscopic SRH. Many patients with massive bleeding were included in their study as compared to our study; therefore a higher incidence of the fluid collection was observed in their study than in our study[19]. In the present study, the identification rate was significantly higher in patients with extravasation or fluid collection than in those with neither (47% vs. 6.6%, p < 0.001). In patients with characteristic imaging findings, the bleeding diverticulum identification rate and subsequent endoscopic hemostasis rate were high. At our hospital, the identification rate was significantly higher for patients who underwent emergency endoscopy within 24 h than in those who underwent elective endoscopy after 24 h (38 vs. 11%, p = 0.001). This is attributed to the higher incidence of extravasation or fluid collection in patients who underwent urgent endoscopy within 24 h (41 vs. 19%, p = 0.014); this is considered a selection bias. We believe that plain CT capable of easily confirming fluid collection as well as contrast-enhanced CT showing images of extravasation are useful for deciding whether urgent endoscopy should be performed in patients with lower gastrointestinal bleeding suspected to be diverticular bleeding.

In the present study, the rate of spontaneous hemostasis following conservative treatment was 87%. The characteristics of patients in whom spontaneous hemostasis was achieved included the absence of extravasation or fluid collection on CT imaging (OR 6.0, p = 0.010), the absence of adhesion of blood on endoscopy (OR 32, p = 0.0012), the absence of diabetes (OR 4.2, p = 0.029), and hemoglobin level ≥10 g/dL (OR 11, p = 0.026). Previous reports have shown a prevalence of 70%–90% for spontaneous hemostasis in patients with diverticular bleeding; however, very few reports describe patients in whom spontaneous hemostasis was achieved[20]. In contrast, re-bleeding was observed during conservative treatment in 16%–38% of the patients with diverticular bleeding[7], and in particular, it has been reported that the rate of re-bleeding is high at 66% in patients with SRH[21]. The rate of re-bleeding is believed to be higher in patients who are men, older (>70 years), severely anemic (hemoglobin levels <8 g/dL), use antithrombotic agents, have a history of hypertension and hyperlipidemia, have a high body mass index, have poor ability to perform activities of daily living[22], present with tachycardia at the time of the consultation, have the bleeding diverticulum located in the ascending colon, and who have a long hospital stay[22–26]. The present results regarding the

characteristics of patients with a high rate of spontaneous hemostasis following conservative treatment are in agreement with the previously reported characteristics of patients with re-bleeding. In Japan, the prevalence of colonic diverticulitis is increasing[27]. Moreover, the frequency of finding diverticula by chance on endoscopy is <10% in individuals aged <40 y in contrast to 50%–66% in elderly individuals aged ≥80 y[28–30]. We believe that in the future, with the aging of society, diverticular bleeding will be a common disease. The comorbidities increase with age, and oral medications, such as aspirin and NSAIDs, will be used more. For elderly individuals, the invasiveness of urgent lower gastrointestinal endoscopy is high; thus, it is extremely important to examine the spontaneous hemostasis rate to reduce unnecessary invasive examinations. For patients without extravasation and fluid collection on CT, without diabetes, and with hemoglobin levels ≥10 g/dL, we believe that urgent endoscopy is not necessary and that elective endoscopic examination can be considered.

The present study was limited in that it was a retrospective study conducted at a single institution, the study population comprised only Japanese patients, and that the study sample was relatively small.

Among patients with diverticular bleeding, for those who exhibit extravasation or fluid collection on CT examination, performing urgent endoscopy should be considered. It is highly likely that spontaneous hemostasis will be achieved in patients without extravasation and fluid collection on CT examination, and elective endoscopy can be selected for these patients.

## Conclusion

In the present study, for patients with lower gastrointestinal bleeding suspected of colonic diverticular bleeding, characteristic CT findings (extravasation and fluid collection) were related to the identification of the bleeding diverticulum. Patients without characteristic CT findings exhibited a high rate of spontaneous hemostasis. CT findings are useful to judge the timing of endoscopy for patients with colonic diverticular bleeding. We believe that our findings will be particularly helpful for determining the need for urgent endoscopy in clinical situations when limited endoscopy staff members are available.

## Acknowledgments

The authors thank all doctors for participating in this survey.

## Author Contributions

**Conceptualization:** Hitomi Takada, Makoto Kadokura, Fumitake Amemiya.

**Data curation:** Hitomi Takada, Makoto Kadokura, Tomoki Yasumura, Hiroki Yoda, Tetsuya Okuwaki, Naoto Imagawa, Naruki Shimamura, Keisuke Tanaka, Fumitake Amemiya.

**Formal analysis:** Hitomi Takada, Makoto Kadokura.

**Methodology:** Hitomi Takada.

**Supervision:** Hitomi Takada, Makoto Kadokura, Fumitake Amemiya, Nobuyuki Enomoto.

**Validation:** Hitomi Takada.

**Visualization:** Hitomi Takada.

**Writing – original draft:** Hitomi Takada.

**Writing – review & editing:** Makoto Kadokura, Fumitake Amemiya, Nobuyuki Enomoto.

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
