## [Decision Letter · Decision Letter 0]

2 Jan 2020

PONE-D-19-28894

Extravasation and fluid collection on computed tomography imaging in patients with colonic diverticular bleeding

PLOS ONE

Dear Dr. Takada,

Thank you for submitting your manuscript to PLOS ONE. After careful consideration, we feel that it has merit but does not fully meet PLOS ONE’s publication criteria as it currently stands. Therefore, we invite you to submit a revised version of the manuscript that addresses the points raised during the review process.

We would appreciate receiving your revised manuscript by Feb 14 2020 11:59PM. To enhance the reproducibility of your results, we recommend that if applicable you deposit your laboratory protocols in protocols.io, where a protocol can be assigned its own identifier (DOI) such that it can be cited independently in the future. For instructions see: http://journals.plos.org/plosone/s/submission-guidelines#loc-laboratory-protocols

We look forward to receiving your revised manuscript.

Kind regards,

Yan Li

Academic Editor

PLOS ONE

Journal Requirements:

2. In the ethics statement in the manuscript and in the online submission form, please provide additional information about the patient records used in your retrospective study.

Specifically, please ensure that you have discussed whether all data were fully anonymized before you accessed them and/or whether the IRB or ethics committee waived the requirement for informed consent.

If patients provided informed written consent to have data from their medical records used in research, please include this information.

4. Your ethics statement must appear in the Methods section of your manuscript. If your ethics statement is written in any section besides the Methods, please move it to the Methods section and delete it from any other section. Please also ensure that your ethics statement is included in your manuscript, as the ethics section of your online submission will not be published alongside your manuscript.

Reviewers' comments:

Reviewer's Responses to Questions

**Comments to the Author**

1. Is the manuscript technically sound, and do the data support the conclusions?

Reviewer #1: Yes

Reviewer #2: Yes

2. Has the statistical analysis been performed appropriately and rigorously? 

Reviewer #1: Yes

Reviewer #2: Yes

3. Have the authors made all data underlying the findings in their manuscript fully available?

Reviewer #1: Yes

Reviewer #2: Yes

4. Is the manuscript presented in an intelligible fashion and written in standard English?

Reviewer #1: Yes

Reviewer #2: Yes

5. Review Comments to the Author

Reviewer #1: In the manuscript entitled ‘Extravasation and fluid collection on computed tomography imaging in patients with colonic diverticular bleeding’, the authors carried out a retrospective study to identify factors associated with identification of colonic diverticular bleeding. They found that extravasation and fluid collection on CT imaging were extracted as factors of identification of colonic diverticular bleeding by colonic endoscopy. Although this study seems well designed and the results are relevant to clinical practice, some issue should be properly addressed.

Comments to authors

Major comments

1. Although authors stated that endoscopy within 24 hours was important in identifying colonic diverticular bleeding, early endoscopy may have been performed in patients suspected of aggressive active bleeding. Therefore, this result may be largely related to bias. The reviewer thinks that the analysis should be based only on background findings such as CT findings as shown in the title.

2. The authors stated that extravasation and fluid collection on CT imaging were extracted as factors of identification of colonic diverticular bleeding by colonic endoscopy. The authors should refer to the characteristics of patients that had bleeding findings in the absence of such findings.

Minor comments

1. The case of diverticular bleeding is stated as 20% in the result, but 21% in the discussion.

2. The similar sentences are repeated in introduction and discussion section. Please organize these sentences.

Reviewer #2: Thank you for the invitation, in this interesting paper, authors described ‘Extravasation and fluid collection on computed tomography imaging in patients with colonic diverticular bleeding’, the paper is sound and has potential for the future clinical application, however, the authors still need to revise it before the final acceptance.

Comments to authors

1. writing skills, please don't cop and paste some sentences, please rewrite,

2. Please list patients characteristics so that the others could better understand the whole story.

3. diverticular bleeding is stated as 20%, but why showed 21% in the discussion part.

6. PLOS authors have the option to publish the peer review history of their article (what does this mean?). If published, this will include your full peer review and any attached files.

Reviewer #1: No

Reviewer #2: No

---

## [Author Response · Author response to Decision Letter 0]

2 Feb 2020

Dear Editor and the reviewers,

 Thank you for your kind review of our manuscript.

 Considering all the suggestions, the manuscript is revised. We believe that our manuscript is now suitable for the publication in the journal.

1) Please update this statement to indicate whether all data were fully anonymized before the authors accessed them and whether the IRB or ethics committee waived the requirement for informed consent.

Thank you for your comment. The institutional review board of Municipal Hospital of Kofu approved the study (ethics committee for clinical studies of Municipal Hospital of Kofu : Rinshoukenkyu-Rinrishinsa-Iinkai (in Japanese), approval number 31-24), the ethics committee waived the requirement for informed consent. All data were fully anonymized before we accessed them. This is now modified in the methods session. (page3, lines 90-94)

2) You note that your data are available within the Supporting Information files, but no such files have been included with your submission. At this time we ask that you please upload your minimal data set as a Supporting Information file, or to a public repository such as Figshare or Dryad.

Thank you for your comment. All data are within the manuscript, and we have no additional supporting files.

---

## [Editor Report · Decision Letter 1]

6 Feb 2020

PONE-D-19-28894R1

Extravasation and fluid collection on computed tomography imaging in patients with colonic diverticular bleeding

PLOS ONE

Dear Dr. Takada,

Thank you for submitting your manuscript to PLOS ONE. After careful consideration, we feel that it has merit but does not fully meet PLOS ONE’s publication criteria as it currently stands. Therefore, we invite you to submit a revised version of the manuscript that addresses the points raised during the review process.

We would appreciate receiving your revised manuscript by Mar 22 2020 11:59PM. To enhance the reproducibility of your results, we recommend that if applicable you deposit your laboratory protocols in protocols.io, where a protocol can be assigned its own identifier (DOI) such that it can be cited independently in the future. For instructions see: http://journals.plos.org/plosone/s/submission-guidelines#loc-laboratory-protocols

We look forward to receiving your revised manuscript.

Kind regards,

Yan Li

Academic Editor

PLOS ONE

Additional Editor Comments (if provided):

Please include a point-to-point rebuttal letter when submitting the revision, also include in the submission system
---

## [Author Response · Author response to Decision Letter 1]

7 Feb 2020

Dear Editor and the reviewers,

 Thank you for your kind review of our manuscript.

 Considering all the suggestions, the manuscript is revised. We believe that our manuscript is now suitable for the publication in the journal.

Point by points

1) Please update this statement to indicate whether all data were fully anonymized before the authors accessed them and whether the IRB or ethics committee waived the requirement for informed consent.

Thank you for your comment. The institutional review board of Municipal Hospital of Kofu approved the study (ethics committee for clinical studies of Municipal Hospital of Kofu : Rinshoukenkyu-Rinrishinsa-Iinkai (in Japanese), approval number 31-24), the ethics committee waived the requirement for informed consent. All data were fully anonymized before we accessed them. This is now modified in the methods session. (page3, lines 90-94)

2) You note that your data are available within the Supporting Information files, but no such files have been included with your submission. At this time we ask that you please upload your minimal data set as a Supporting Information file, or to a public repository such as Figshare or Dryad.

Thank you for your comment. All data are within the manuscript, and we have no additional supporting files.

3) Additional Editor Comments (if provided):

Please include a point-to-point rebuttal letter when submitting the revision, also include in the submission system.

Thank you for your comment. We fixed the point-to-point rebuttal letter.

---

## [Editor Report · Decision Letter 2]

19 Feb 2020

Extravasation and fluid collection on computed tomography imaging in patients with colonic diverticular bleeding

PONE-D-19-28894R2

Dear Dr. Takada,

We are pleased to inform you that your manuscript has been judged scientifically suitable for publication and will be formally accepted for publication once it complies with all outstanding technical requirements.

With kind regards,

Yan Li

Academic Editor

PLOS ONE
---

## [Editor Report · Acceptance letter]

27 Mar 2020

PONE-D-19-28894R2 

Extravasation and fluid collection on computed tomography imaging in patients with colonic diverticular bleeding 

Dear Dr. Takada:

I am pleased to inform you that your manuscript has been deemed suitable for publication in PLOS ONE. Congratulations! Your manuscript is now with our production department. 

With kind regards,

on behalf of

Dr. Yan Li 

Academic Editor

PLOS ONE